

# Occurrence of Nosemosis in honey bee, *Apis mellifera* L. at the apiaries of East Kazakhstan

Abdrakhman Baigazanov[1,2,*], Yelena Tikhomirova[1,*], Natalya Valitova[3], Maral Nurkenova[1,2], Ainur Koigeldinova[2], Elmira Abdullina[1,2], Olga Zaikovskaya[2], Nurgul Ikimbayeva[2], Dinara Zainettinova[2] and Lyailya Bauzhanova[4]

[1] Department of Veterinary, Faculty of Veterinary Medicine and Agricultural Management, Shakarim University, Semey, East Kazakhstan Region, Kazakhstan

[2] Agrotechnopark Scientific Center, Veterinary and Food Safety Laboratory, Shakarim University, Semey, East Kazakstan Region, Kazakhstan

[3] School of Earth and Environmental Sciences, D. Serikbayev East Kazakhstan Technical University, Ust-Kamenogorsk, East Kazakhstan Region, Kazakhstan

[4] Department of Zoo Technology, Genetics and Breeding, Toraighyrov University, Pavlodar, Pavlodar Region, Kazakhstan

[*] These authors contributed equally to this work.

Corresponding authors
Yelena Tikhomirova,
tihomirova.82@mail.ru
Natalya Valitova, valitova-n@mail.ru

## ABSTRACT

Nosemosis is the most common disease in honey bee *Apis mellifera* L., and is a major issue related to bee health worldwide. Therefore, the purpose of this research study was to determine prevalence of microsporidia parasitic infection of the genus *Nosema* spp. in East Kazakhstan Region (EKR). In the years of 2018 –2021, 394 honey bee samples were collected at 30 apiaries located in four districts of East Kazakhstan Region (Katon-Karagay, Urzhar, Borodulikhinsky, and Shemonaikhinsky). In order to determine the level of infestation, firstly, the presence of *Nosema* spp. spores was detected using optical microscopy, and then the average amount of spores per bee was counted using a hemocytometer. The degree of nosemosis prevalence was determined in points by means of a semi-quantitative method, and as a percentage from the total of samples and of the amount of positive tests. At the outcome of the study, microsporidia of the genus *Nosema* spp. were detected in 23.3% of cases (92 samples). Prevalence at its low degree was found in six samples (1.5%), at an average degree in 55 samples (14%), and at a high one in 31 samples (7.9%). This research study proved that microsporidia of the genus *Nosema* spp. are widely spread at the apiaries of East Kazakhstan Region in different orographic and climatic conditions. Notwithstanding that it was impossible to statistically determine any significant differences between the dependence of nosemosis prevalence and the apiary location, this indicator is actually higher in the mountainous regions than in the steppe. Concurrently, a close inverse correlation was recognized between the amount of spores in one bee and the level of infestation in bee families from the duration of the vegetation season at the apiary location. This gives grounds to assert that the environmental factors have an impact on formation and development of nosemosis. The results of the research presented in the article indicate the need for further research aimed at increasing the number of studied apiaries, and above all the use of molecular biology methods to distinguish the species that cause nosemosis infection (PCR).

## INTRODUCTION

The honey bee is the most important insect-pollinator of agricultural crops worldwide, pollinating more than 90% of all flowering plants, and is of particular scientific and practical interest as a resource species (*Food and Agriculture Organization of the United Nations , FAO*; *Botías et al., 2013*; *Kucher et al., 2016*; *Chen et al., 2013*; *Cridl, Tsutsui & Ramírez, 2017*; *Danner et al., 2017*; *Doublet et al., 2017*; *Ostroverhova, 2018*; *Rodriguez García et al., 2018*). In addition, the honey bee is an asset of economic value owing to the selection of beekeeping products used by people for nutrition and medical treatment (*Sforcin, Bankova & Kuropatnicki, 2017*; *Pasupuleti et al., 2017*; *Shumkova et al., 2018*). The study of the honey bee *Apis mellifera* L. (Hymenoptera: Apidae) being an ecologically and economically significant species, represents a topical and essential research study worldwide (*Krivcov & Lebedev, 2011*; *Kucher et al., 2016*; *Tlak Gajger et al., 2015*; *Ostroverhova, 2018*).

Production of honey and beekeeping products in the East Kazakhstan Region is a long-standing tradition based on diverse and rich vegetation suitable for honey production, as well as because of the presence of favourable natural, climatic and ecological conditions. The East Kazakhstan Region is not only the birthplace of apiculture for Kazakhstan, but is also the main exporter of honey in the Republic (*Rib, 2004*; *Miheeva, 2016*).

It is known that various pathogens and honey bee parasites produce an adverse impact on bee families' lifespan. Additionally, they are considered to be among the key factors causing global deaths of honey bee families (*Murilhas, 2002*; *Higes, Martín-Hernández & Meana, 2010*; *Evans & Schwarz, 2011*; *Botías et al., 2013*; *Tlak Gajger et al., 2015*; *Doublet et al., 2017*; *Higes et al., 2020*; *MacInnis, Keddie & Pernal, 2021*; *Marín-García et al., 2022*).

Some of the most dangerous and widespread parasites on adult honey bees are microsporidia of the genus *Nosema* spp. (Nosematidae), parasitizing in the epithelial cells of the mid-gut (*Bailey, 1981*; *Matheson, 1993*; *Chernyshev, 2012*; *Rangel et al., 2015*; *Golubeva, 2018*; *Jabal-Uriel et al., 2022a*; *Jabal-Uriel et al., 2022b*). To date, there have been identified several species of microsporidia that affect honey bees, *i.e.*, *Nosema apis* (*Zander, 1909*) and *Nosema ceranae* (*Fries et al., 1996*). Clinical manifestations of *Nosema* spp. infection vary by pathogen, at individual and colonial levels alike. For instance, an entirely different epidemiology attributes to *Nosema ceranae*, and consistently occurs at the hives throughout the year, which, therefore, leads to reduction and destruction of the colony. On the contrary, *Nosema apis* takes a seasonal pattern with a typical peak of its occurrence and effect on bee families in winter, causing diarrhoea, the presence of which is noticeable by way of defecated hive frames, honeycombs, outer walls of the hive, *etc.* (*Higes et al., 2005*; *Paxton et al., 2007*; *Il'yasov et al., 2013*; *Dar & Sheikh Bilal, 2013*; *Zinatullina, 2016*; *Zinatullina, Domatskaya & Domatskiy, 2017*; *Doublet et al., 2017*; *Tokarev et al., 2018*; *Fleites-Ayil, Quezada-Euán & Medina-Medina, 2018*; *Li et al., 2019*; *Higes et al., 2020*).
Unlike *Nosema apis*, the large-scale pathogenic effect of *Nosema ceranae* on the honey bee body system was recorded (*Martin-Herna'ndez et al., 2007*; *Higes, Martín-Hernández & Meana, 2010*; *Golubeva, 2018*). The conclusions were drawn on localization of the *Nosema ceranae* spores in various organs and tissues of honey bee, for instance, in malpighian tubule system, fat body, hypopharyngeal and salivary glands, *etc.* (*Chen et al., 2009*; *Gisder et al., 2010*; *Golubeva, 2018*). However, there is recent evidence of high tropism in two species of *Nosema* spp. to the cells of ventricular epithelium. Thus, both of these microsporidia can negatively affect the life expectancy of individuals in a bee colony, disrupting age-related polyethism with an intensity reflecting their prevalence in the colony (*Higes et al., 2020*).

Very recently, a new parasite, *Nosema neumanni* (*Chemurot et al., 2017*), was found in the Ugandan bees of, thus far, unknown occurrence and undescribed impact on *A.mellifera* L. (*Giovanni et al., 2018*; *Mazur & Gajda, 2022*).

*Nosema apis* and *Nosema ceranae* were found on all continents where *A. mellifera* L. is present. For instance, in Africa (*Fries et al., 2003*; *Higes et al., 2009*), in Europe (*Higes, Martín & Meana, 2006*; *Paxton et al., 2007*; *Klee et al., 2007*; *Zinatullina et al., 2011*; *Zinatullina et al., 2018*; *Chernyshev, 2012*; *Golubeva, 2018*; *Ostroverkhova, 2020*), in the Americas (*Chen et al., 2008*; *Calderón et al., 2008*), and in Asia (*Chen et al., 2009*; *Shirzadi & Razmi, 2021*).

The first case of infestation of bees by microsporidia of the genus *Nosema* spp. in Kazakhstan was recorded in 2012 in Ulan District (the village of Saratovka) of East Kazakhstan Region (*Ospanova et al., 2012*), where a number of apiaries died out, as well as in Katon-Karagay Region in 2015 (*Valitova, 2017*). Moreover, it was possible to witness an acute type of nosemosis: the infected bees died in a short span of time, and the defecated inner walls of the hive were observed, as well as the death of queens, a great amount of the dead bees nearby the entrances and at the bottom of the hive. Owing to the studies carried out by means of scanning electron and light microscopy, the oval, well-translucent spores of an intracellular parasite of the genus *Nosema spp.* were detected in the mid-guts of bees (*Ospanova et al., 2012*; *Valitova, 2017*). Still, due to the lack of appropriate methods to date of the research study, the diagnostic analysis of the pathogen species did not take place.

It does not seem possible to provide an impartial assessment of the epizootic situation as regards nosemosis, since systematic monitoring of this disease in East Kazakhstan Region did not take place.

In light of the fact, that many studies prove negative effects of nosemosis at the level of a bee family (*Farrar, 1947*; *Fries, 1998*; *Anderson & Giacon, 1992*; *Higes et al., 2009*; *Mayack & Naug, 2009*; *Botías et al., 2013*; *Tlak Gajger et al., 2015*; *Ostroverhova, 2018*; *Golubeva, 2018*; *Bajgazanov, Pashayan & Tikhomirova, 2019*; *Bajgazanov & Tikhomirova, 2020*; *Kartal et al., 2021*; *Salkova et al., 2022*; *Houdelet et al., 2022*), the need arises to proceed with monitoring studies of honey bee at the apiaries of East Kazakhstan Region.

The relevance of research into microsporidian parasitic infections in honey bee is associated with the development trend in beekeeping in the Republic of Kazakhstan, where, as of 2020, the total of bee families reached 133,672 (*Byuro Nacionalnoj statistiki Agenstva po strategicheskomu planirovaniyui reformam Respubliki Kazakhstan, 2021*). Of

them, 71% or 94,421 bee families are concentrated in East Kazakhstan Region and produce up to 80% of the total honey of the Republic annually.

In connection therewith, the purpose of this research was to study infestation of honey bee by microsporidia of the genus *Nosema* spp. at the apiaries of different districts in the East Kazakhstan Region. Species-level identification of *Nosema* spp. did not take place at that stage.

## MATERIALS & METHODS

### Research sites

The research was carried out at Agrotechnopark Scientific Center, at Veterinary and Food Safety Laboratory of Shakarim State University of Semey. The apiaries of the East Kazakhstan Region of the Republic of Kazakhstan were used for sampling.

The area of sampling and sample testing is located in the northeastern and southern parts of the Region. The lowlands and generally flat plains are typical for the Districts of Borodulikhinsky and Shemonaikhinsky, while Katon-Karagay and Urzhar Districts are located in the mountains.

The climate of the studied districts of East Kazakhstan Region is distinctly continental, defined by long severe winters and dry, rather hot and torrid summers, with large temperature fluctuations both according to the year seasons and during the day. Continentality diminishes slightly in the mountainous areas and at the foothills.

The climatic conditions of the lowlands and mountainous areas differ from each other in terms of both temperature and amount of precipitation (*Shcherbakov, Shcherbakova & Kotuhov, 1991*; *Rib, 2004*; *Nurtazin, 2017*; *Krutova, 2021*; *Weather Spark, 2022*).

In the Katon-Karagay Region, winters are long and relatively cold. Cool and short summers (90 to 100 days) precondition brevity of the vegetation season.

The climate in Urzhar Region is defined as semi-dry, mildly hot.

The Borodulikhinsky District is located in the dry steppe subzone of the steppe zone.

The Shemonaikhinsky District belongs to the mountain meadow steppe and foothill steppe, *i.e.*, moderately humid natural and climatic zones.

The vegetation season in Katon-Karagay, Borodulikhinsky, Shemonaikhinsky and Urzhar Districts lasts 100, 122, 126 and 166 days respectively.

Figures 1A and 1B shows the research sites.

### Sampling

The samples (*A. mellifera* L.) were collected from 394 bee colonies from 30 apiaries located in four districts of East Kazakhstan Region (Katon-Karagay, Urzhar, Borodulikhinsky, Shemonaikhinsky) in the spring (April–May) from 2018 to 2021.

For the purposes of this research, adult living bees were selected (in the amount of at least 10% of the total of bee families at each apiary), as well as the dead bees were collected near the hives. The samples were taken randomly at the hive entrance or on the hive frames away from the brood nest. Each sample contained at least 50 units for the test study. Samples of the living bees were dusted down into a plastic envelope with free access for air, and into paper bags. The dead bees was collected into separate paper bags. Each sample

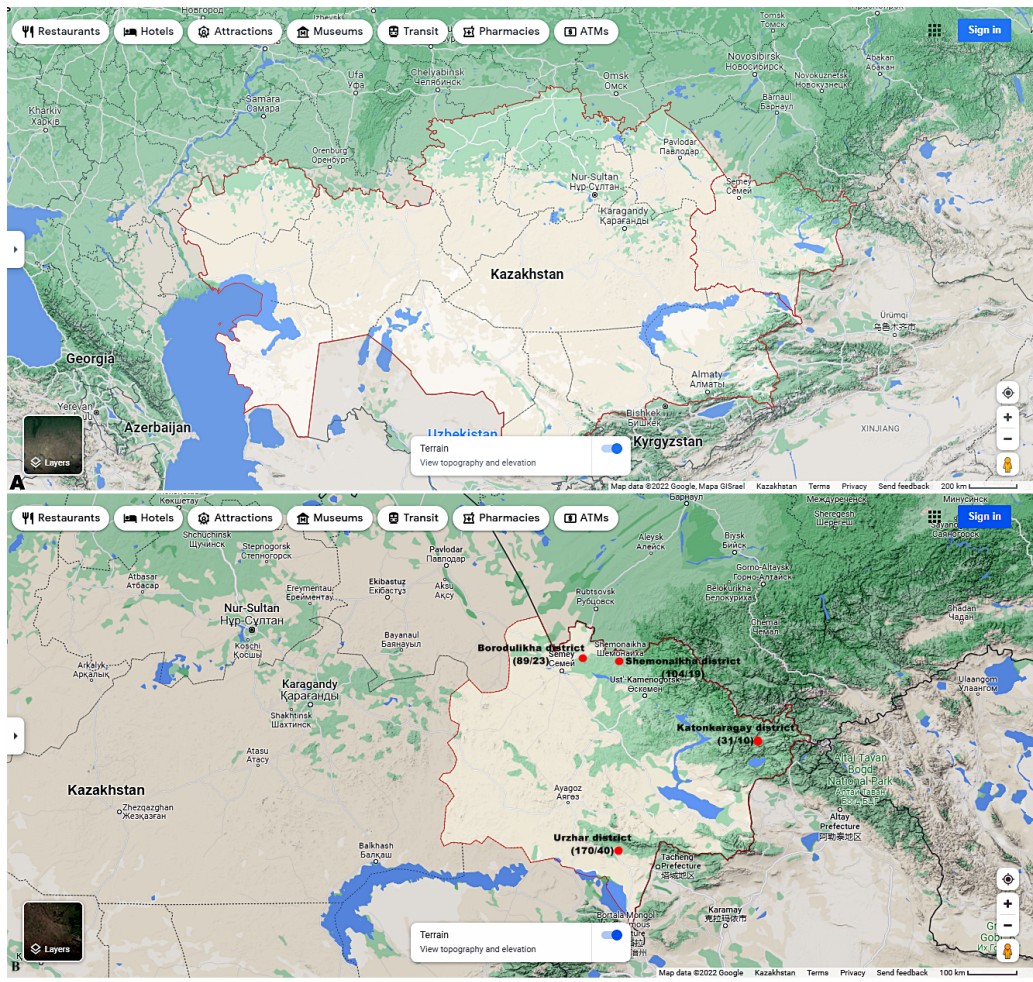

**Figure 1** (A) Map of Kazakhstan and location of EKR. (B) Map of sampling sites in EKR. The ratio of the total of samples/amount of samples infected with microsporidia of the genus *Nosema spp.* is shown in brackets. Map data©2022 Google.

was numbered. The bag was marked with the beehive number, sample number, and the date of sampling.

Prior to laboratory testing, live bees were frozen for immobilization at −20 °C for 15–20 min by following the protocol set for Nosemosis diagnose in various studies (*Antonov et al., 1987*; *Grobov, Smirnov & Popov, 1987*; *Topolska & Hartwig, 2005*; *Il'yasov et al., 2013*; *Pohorecka et al., 2018*; *Shumkova et al., 2018*).

### Research methods

The analysis of infestation of honey bees (*A. mellifera* L.) with microsporidia was carried out in accordance with the ''Methodological guidelines for laboratory studies on honey bee nosematosis'' in force on the territory of the Republic of Kazakhstan, approved on April 25, 1985 (*Antonov et al., 1987*).

Microscopic counting of the bees infected with the causative agents of nosemosis was done by means of a Micmed-5 XC1450 optical microscope as well as a hemocytometer. The hemocytometer counted the amount of nosema spores and was used to count the average amount of spores per bee (*Zinatullina, 2018*).

## Method of optical microscopy

The bee samples were tested by a group method. In order to accomplish this, the mid-gut was removed from the frozen units, and the abdomen was taken from the dead bees. Those were then placed in a porcelain mortar with a pestle; distilled water was added at the rate of 1 ml per bee tested. The mortar contents were thoroughly pulverized until they became streak free. A drop of the prepared suspension was applied to the slide plate, and then covered with a cover glass so that no air bubbles remained. Excess water was removed by filter paper. Further-on, microscopy was conducted at a mean magnification (x400) in a slightly darkened field of vision. At least 20 fields of vision were viewed (*Grobov, Smirnov & Popov, 1987*; *Voronin & Issi, 1974*).

In the case of a positive test, in the microscope field of vision there were detected oval, slightly curved, sometimes straight or rice-grain shaped spores of the genus *Nosema* spp., strongly refracting the light and with the dimensions of $4.5-7.5 \times 2.0-3.5$ μm as shown in Fig. 2 (*Zander & Böttcher, 1984*; *Fries et al., 1996*).

## Method of counting the amount of spores and estimation of the level of infestation

In order to proceed with quantitative diagnostics of nosemosis, the spores were counted in the hemocytometer of the Goryaev counting-chamber device. The amount of nosema spores was counted in five large squares divided diagonally into 16 small hemocytometer grids. Then, by multiplying by 50.000, it was possible to obtain the amount of spores in 1 ml of suspension, or per bee. The level of infestation was assessed on a 4-point scale in view of the amount of spores per bee (Table 1) (*Zinatullina et al., 2018*; *Zinatullina, 2018*).

## Statistical method

Systematization of the source data, statistical processing and analysis of the results were accomplished by means of Microsoft Office Excel 2016 spreadsheets. Statistical analyses were also performed using the online calculators available at https://medstatistic.ru (*Marapov, 2013*). Reliability of the difference in the obtained data was determined by the Student's $t$-test. Analysis of the total of apiaries and bee families affected by nosemosis and the degree of prevalence by area was defined using the Student's $t$-test method when comparing the relative values.

Statistical estimates of the extent of bee family infestation and the amount of spores per bee were done using the Student's $t$-test method when comparing the mean values for independent variables (*Tokarev et al., 2018*). For this reason, the following key indicators were calculated: mean value, variance, quadratic deviation, standard error, $t$-test, and significance point. The correlation ratio (r) was calculated in order to define dependence between the amount of spores per bee and the degree of impact upon bee families from the duration of the vegetation season.

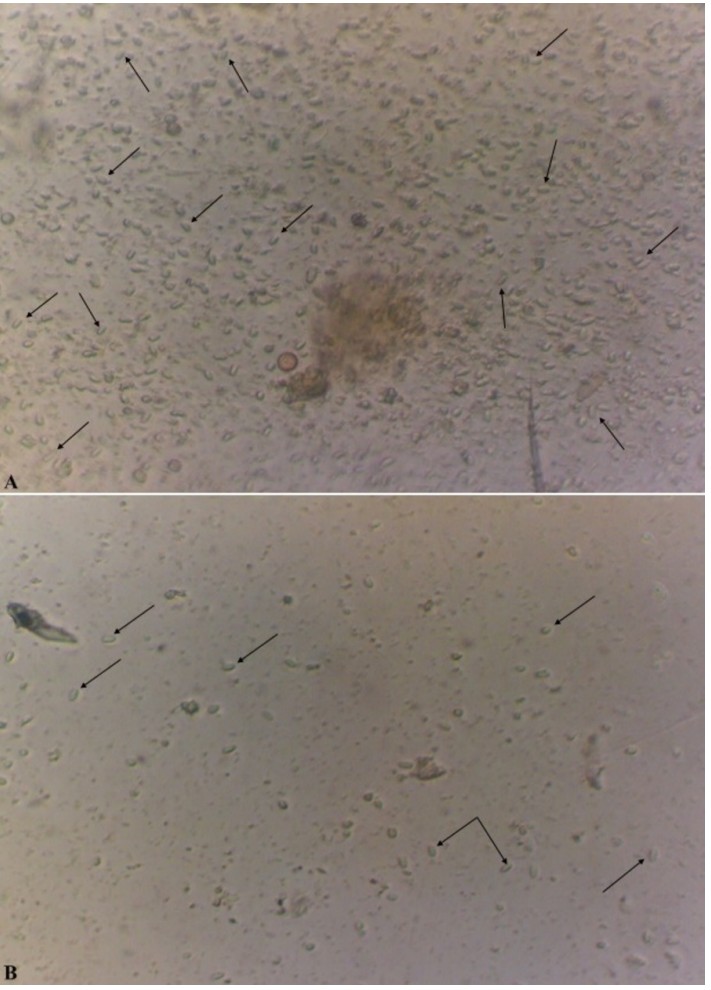

**Figure 2 Spores of microsporidia of the genus *Nosema spp* (x400).** The level of infestation of honey bee: a high level of infestation (three points) is shown in (A); a low level of infestation (one point) is shown in (B).

**Table 1 Assessment of infestation of the honey bee families by microsporidia of the genus *Nosema spp*. (*Zinatullina, 2018*).**

| Level of infestation, points | 1–low | 2–average | 3–high | 4–severe |
|---|---|---|---|---|
| Amount of spores per bee, million | up to 5 | from 5.05 to 25 | from 25.05 to 75 | over 75 |

## RESULTS

### Analysis of microscopic examination

For the purposes of determination of prevalence of nosemosis in East Kazakhstan region, the presence of the *Nosema* spp. spores was initially detected in the bee samples employing the light microscopy, and then a haemocytometer was used to count the number of

**Table 2** Infestation of apiaries by microsporidia of the genus *Nosema spp.* in different districts of the East Kazakhstan Region.

| Districts of East Kazakhstan Region | Apiaries | | | Total amount of bee families at the apiaries under study, units |
|---|---|---|---|---|
| | Total examined, colonies | Infestation by nosemosis, colonies | Prevalence, % | |
| Katon-Karagay | 6 | 5 | 83.3 | 320 |
| Urzhar | 10 | 8 | 80 | 1,430 |
| Borodulikhinsky | 8 | 5 | 62.5 | 892 |
| Shemonaikhinsky | 6 | 3 | 50 | 1,050 |
| In the Region: | 30 | 21 | 70.0 | 3,692 |

microsporidia per bee. Species-level identification of *Nosema* spp. did not take place at that stage.

Microscopic analysis indicated that nosemosis was widespread at the apiaries of the East Kazakhstan Region. Out of 30 apiaries under study, with the total of 3,692 bee families, nosemosis was detected at 21 apiaries (70%) (Table 2).

According to the results (Table 2), it was concluded that nosemosis infestation was higher in apiaries of mountainous regions (Katon-Karagay and Urzhar) as compared to the regions with steppe slope (Borodulikhinsky and Shemonaikhinsky). In the mountainous regions, infestation of the apiaries by nosemosis reached 83.3 and 80% (81.65% on average), and in the steppe regions 62.5 and 50% (56.25% on average). The comparative analysis given in Fig. 3 shows that there are no statistically significant differences between the amount (proportion) of the apiaries affected by nosemosis and their location ($p > 0.05$).

## Levels of infestation of bee families

Out of 394 honey bee samples under study, the spores of the microsporidia genus *Nosema* spp. were found in 92 samples, or in 23.3% of the cases.

At that, different levels of infestation of honey bee were outlined (in percentage of the total amount of samples/the amount of positive tests). As for the regional average, a low degree of prevalence was detected in six samples (1.5/6.5%), average in 55 samples (14.0/59.8%), and a high one in 31 samples (7.9/33.7%).

Consequently, infestation of bee families in different districts of East Kazakhstan Region ranged from weak (one point) to strong (three points). There was no particularly severe infestation (four points) of the bees (Table 3).

Low level of infestation was not detected among bee families of district Katon-Karagay district according to the results given in Table 3. A high degree of prevalence (3 points) was recorded for the Katon-Karagay (60% of positive tests) and Borodulikhinsky (52.2%) Districts. In the Urzhar and Shemonaikhinsky Districts, the average degree of prevalence prevails (70.0 and 73.7% respectively out of the positive tests).

When comparing the two districts (Fig. 4), difference in the incidence of bee families with nosemosis remains statistically unreliable ($p > 0.05$).

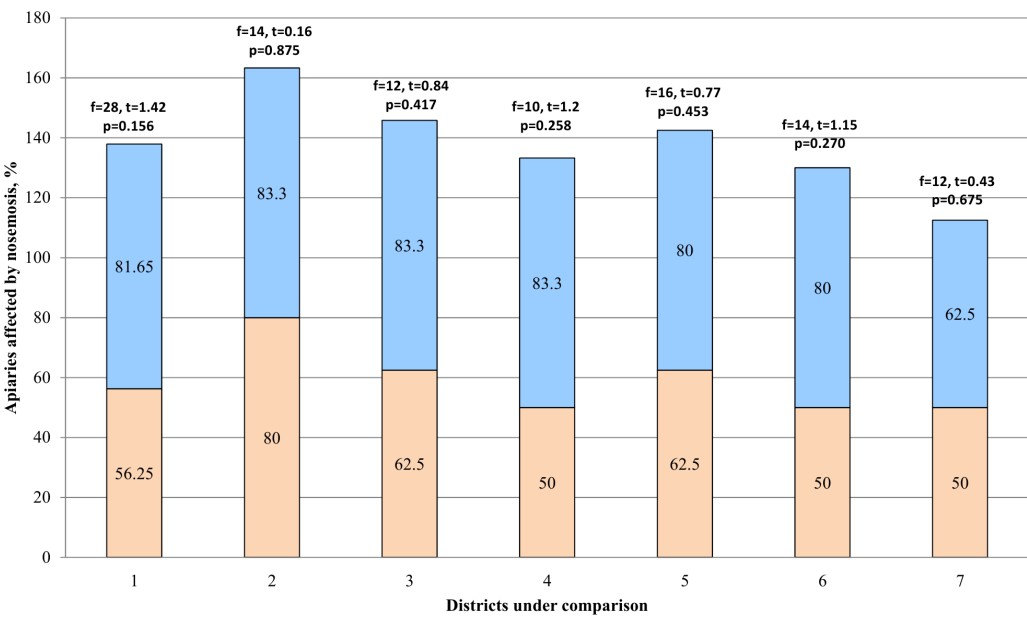

**Figure 3  Comparative analysis of the apiaries affected by nosemosis in the districts.** 1–mountainous/steppe; 2–Katon-Karagay/Urzhar; 3 –Katon-Karagay/Borodulikhinsky; 4–Katon-Karagay/Shemonaikhinsky; 5–Urzhar/Borodulikhinsky; 6–Urzhar/Shemonaikhinsky; 7–Borodulikhinsky/Shemonaikhinsky.

**Table 3  Levels of infestation of bee families.**

| Districts of East Kazakhstan Region | Samples | | | Level of infestation (amount of samples, cases/% from the total amount of samples (% from the amount of positive tests)) | | | | | Amount of spores per bee, million |
|---|---|---|---|---|---|---|---|---|---|
| | Tested, cases | Positive | | – | 1 point | 2 points | 3 points | 4 points | |
| | | cases | % | | | | | | |
| Katon-Karagay | 31 | 10 | 32.2 | 21/67.5 | 0/0 (0) | 4/12.9 (40) | 6/19.4 (60) | 0 | $22.6 \pm 2.76$ |
| Urzhar | 170 | 40 | 23.5 | 130/76.5 | 3/1.8 (7.5) | 28/16.5 (70.0) | 9/5.3 (22.5) | 0 | $14.83 \pm 1.25$ |
| Borodulikhinsky | 89 | 23 | 25.8 | 66/74.16 | 2/2.2 (8.7) | 9/10.1 (39.1) | 12/13.5 (52.2) | 0 | $20.9 \pm 2.17$ |
| Shemonaikhinsky | 104 | 19 | 18.3 | 85/81.7 | 1/1 (5.2) | 14/13.5 (73.7) | 4/3.8 (21.1) | 0 | $13.5 \pm 1.85$ |
| Total (mean value*) | 394 | 92 | 23.4* | 302/76.6 | 6/1.5 (6.5) | 55/14.0 (59.8) | 31/7.9 (33.7) | 0 | 17.96* |

In the Katon-Karagay and Borodulikhinsky Districts, the amount of spores per bee is significantly higher than in Urzhar ($f = 48, t = 2.56, p = 0.011; f = 27, t = 2.74, p = 0.006$) and Shemonaikhinsky ($f = 61, t = 2.436, p = 0.015; f = 40, t = 2.6, p = 0.009$).

The statistically significant differences were recorded with regard to the amount of spores per bee *versus* the duration of the vegetation season in the area ($p > 0.05$). The

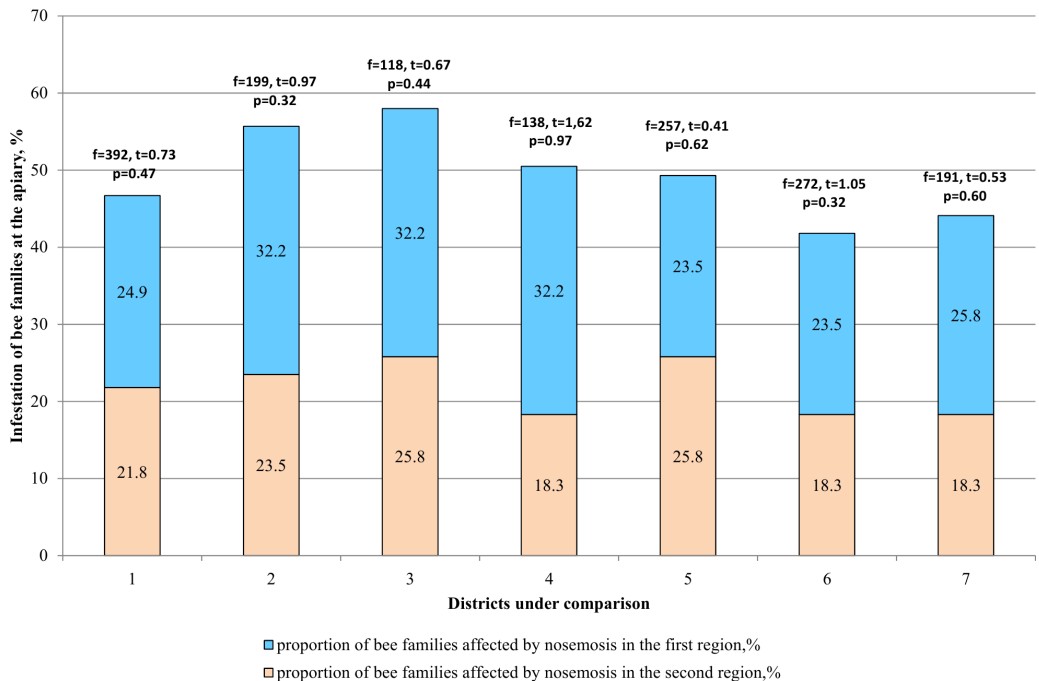

**Figure 4  Comparative analysis of the level of infestation of bee families by nosemosis at the apiaries.** 1–mountainous/steppe; 2–Katon-Karagay/Urzhar; 3 –Katon-Karagay/Borodulikhinsky; 4–Katon-Karagay/Shemonaikhinsky; 5–Urzhar/Borodulikhinsky; 6–Urzhar/Shemonaikhinsky; 7–Borodulikhinsky/Shemonaikhinsky.

vegetation season in the Katon-Karagay, Borodulikhinsky, Shemonaikhinsky and Urzhar Districts lasts 100, 122, 126 and 166 days, and the amount of spores per bee is 22.6 ± 2.76, 20.9 ± 2.17, 13.5 ± 1.85 and 14.83 ± 1.25 million units respectively (Fig. 5). The correlation coefficient (r) is −0.720. The relationship between the studied criteria is inverse. The tightness (strength) according to the Cheddock's scale is strong.

Consequently, it is possible to ascertain reliably that with reduction in the duration of the vegetation season, the amount of spores per bee and the degree of incidence of nosemosis in bee families increases.

## DISCUSSION

The outcome of our research studies proved the presence of microsporidia of the genus *Nosema* spp. in 23.3% of the analysed bee samples taken from four districts of the East Kazakhstan region.

The highest degree of prevalence was recorded in the Katon-Karagay District. Of 31 samples tested, 10 (32.2%) were positive (Table 2).

In the Urzhar District positive samples of *Nosema* spp. were found in 40 out of 170 samples under study (23%).

The lowest level of infestation was recorded in the samples of honey bee in the Shemonaikhinsky District. Out of 104 studied samples, only 19 (18.3%) revealed

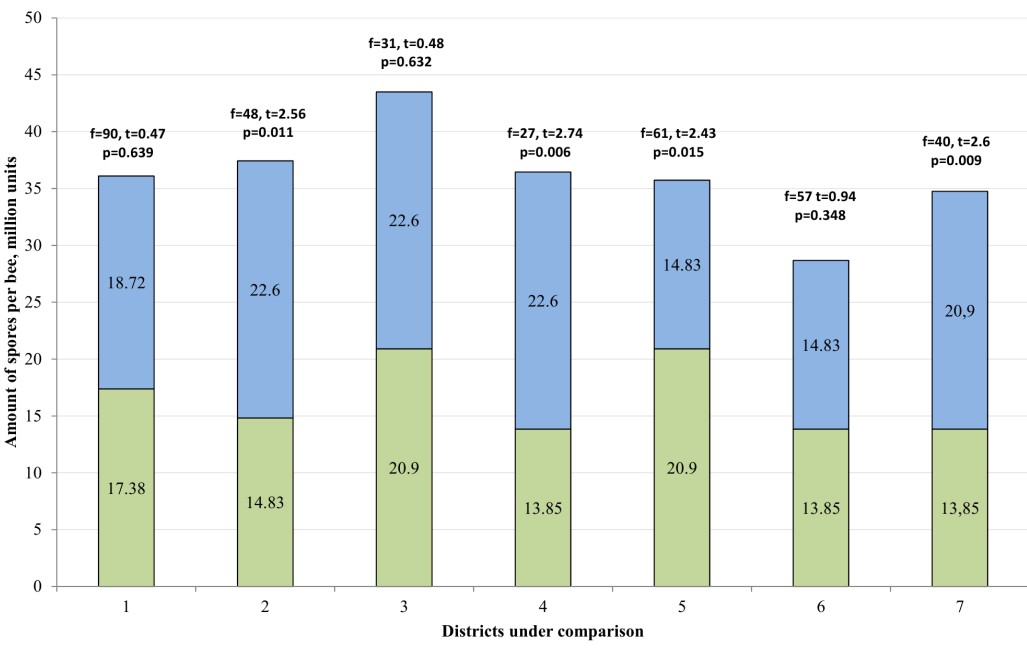

**Figure 5** **Comparative analysis of the amount of spores per bee.** 1–mountainous/steppe; 2–Katon-Karagay/Urzhar; 3 –Katon-Karagay/Borodulikhinsky; 4–Katon-Karagay/Shemonaikhinsky; 5–Urzhar/Borodulikhinsky; 6–Urzhar/Shemonaikhinsky; 7–Borodulikhinsky/Shemonaikhinsky.

microsporidia of the genus *Nosema* spp. Nonetheless, in Borodulikhinsky District of the same part of the Region, the level of infestation reached 25.8%.

Our data show that this type of infection is widespread at the apiaries. Our findings align with the data available in other countries. For instance, the analysis of the study results of the bee families representing 18 regions of Russia (Arkhangelsk, Belgorod, Voronezh, Kirov, Leningrad, Moscow, Orenburg, Penza, Tomsk, Tula, Tyumen, Altai Republic, Krasnodar, Perm and Stavropol Territories, Republics of Mari El, Tatarstan, Udmurtia) shows that the apiaries in question are affected by nosemosis (*Zinatullina et al., 2018*).

The studies that took place in a number of European countries inclusive of Ukraine, Spain, France, Germany, Switzerland, Denmark, Finland, Greece, Hungary, the Netherlands, the United Kingdom, Italy, Serbia, Poland, Slovenia, Bosnia and Herzegovina, Sweden, *etc.* (*Zinatullina et al., 2018*) endorse the distribution of nosemosis in these countries.

In recent times, only *Nosema cerana* has been detected in many countries. For instance, in the course of the survey research in Hatay province it was found that, in overall, 20% of the total count of hives were infected. At that point, no *Nosema apis* has been detected (*Aykut, Mehmet & Bilal, 2022*).

The research studies conducted by many scientists proved that microsporidia parasitic infection is predetermined by a number of factors, inclusive of temperature, seasonal patterns, nutrient availability, presence of micro-organisms or absence thereof, as well as

the stage of development, sex and genetics of the bearer (resistance to microsporidia), *etc.* (*Willis & Reinke, 2022*).

It is evident that the biology of *Nosema* spp. depends on temperature. In fact, development of *Nosema apis* demonstrates a seasonal pattern, in which the level of infestation reaches its peak in spring, decreases during the summer time, and is followed by a repeated lower peak in autumn, after which it again decreases in winter (*Traver, Williams & Fell, 2012*). *Nosema ceranae* can be detected all year round; it develops more rigorously, particularly in the time frame from April to June. In the recent times, this species of *Nosema* spp. occurs more frequently, and infestation of bee families with it is always higher than that with *Nosema apis* (*Traver, Williams & Fell, 2012*; *Higes et al., 2013*; *Aykut, Mehmet & Bilal, 2022*). This may be owing to the fact that *Nosema ceranae* adapts to temperature fluctuations better (*Martín-Hernández et al., 2009*). Differences in the number of causative agents as observed in between spring and autumn may equally be related to the size of a bee family, as well as physiological features of bees in these seasons. Older bees have a higher level of contamination with *Nosema ceranae* (*Jabal-Uriel et al., 2022a*; *Jabal-Uriel et al., 2022b*).

The level of infestation in bee families at the same apiary over the summer time may differ not least because of the field bees from different bee families choosing different plants. Chemical composition of honey and pollen collected from different plants may vary (*Jabal-Uriel et al., 2022a*; *Jabal-Uriel et al., 2022b*). Consequently, their impact on bee health will be different as well. For instance, high-quality and diverse pollen nutrition improves the survival rate of the healthy bees and the bees infected with *Nosema ceranae*, whilst the quality of pollen (as reflected through protein content and antioxidative activity) highly impacts the exposure of bees to infection (*Martín-Hernández et al., 2018*; *Jabal-Uriel et al., 2022a*; *Jabal-Uriel et al., 2022b*). The pollen and honey of some plants (manuka) may trigger premature death of bee families (*Malone, Gatehouse & Tregidga, 2001*).

Combined infection and bee pests entail weakened immunity and, most often, result in more severe infestation of the bee families with nosemosis, as well as their premature death (*Martín-Hernández et al., 2018*).

Occurrence of nosemosis is also impacted by the methods of beekeeping, beekeeper competence, and compliance with sanitary measures at the apiary (*Martín-Hernández et al., 2018*). In fact, the burden of disease at the apiaries employing the same beekeeping tools for different hives is significantly higher than that at the apiaries that do not employ the same tools for different hives (*Aykut, Mehmet & Bilal, 2022*). When it comes to isolating the queen and then replacing her with a younger one, the proportion of bees infected with nosemosis gets reduced, maintaining the overall infestation at the level compatible with viability of the colony (*Martín-Hernández et al., 2018*).

Some authors proved, that the endemic bees get infested with nosemosis not as often as the introduced bee races and breeds (*Aykut, Mehmet & Bilal, 2022*).

In addition, it is known that occurrence of nosemosis is often driven by the conditions related to the territorial distribution of bee families.

Adverse environmental conditions (polluted environment, use of pesticide) at the beekeeping sites cause a more severe disease and significantly higher rates of mortality in bees (*Martín-Hernández et al., 2018*).

There is also a point of view that warm climatic conditions are favourable for distribution of *Nosema ceranae* while *Nosema apis* develops in colder climates (*Gisder et al., 2010*; *Higes et al., 2013*; *Shumkova et al., 2018*).

In the course of the study of 28 samples of *A. mellifera* L., typical for three natural and geographical zones of Bulgaria (Southern, Northern and Western Bulgaria), it was found that the highest level of infestation (77.2%) by *Nosema ceranae* was attributable to the bees from the northern part of the country. For the bees from the mountainous regions (the Rhodopes, Southern Bulgaria) it was only 13.9% (*Shumkova et al., 2018*; *Shumkova et al., 2020*).

This correlates well with other results confirming that the longer the cold period lasts, the more likely it is for nosemosis to develop, and the greater the degree of infestation in bee families with microsporidia (*Chen et al., 2012*; *Pacini et al., 2021*). Given that duration of the period with the average monthly air temperature below zero is longer in the northern than in the southern regions, the level of infestation with nosemosis can be respectively higher.

The outcome of our research studies showed, that in the Katon-Karagay District, located in the northeastern part of the East Kazakhstan Region in the mountainous area (the vegetation season does not exceed 100 days), with regard to all considered prevalence rates, there were recorded the highest values of nosemosis in the District, namely 83.3% of all the apiaries under study. In 60% of cases, a high level of infestation of bee families was recorded (three points), as well as the greatest amount of spores per bee (22.9 million).

In the Urzhar Region (mountain terrain, located much to the south of the Katon-Karagay Region, with the vegetation season lasting 166 days), the average level of infestation of bee families prevails (two points in 70% of the cases) at 80% of the apiaries affected by nosemosis, and the amount of spores per bee reaches 14.83 million. This is respectively only 1.33% higher and 6.07% lower than at the apiaries located in the lowlands.

In the steppe (Borodulikhinsky and Shemonaikhinsky Districts), where the vegetation season lasts 122 and 126 days, the climatic conditions are similar, still the rates of occurrence differ. In the Borodulikhinsky District, the total of the apiaries and bee families affected by nosemosis, a high level of infestation of bee families and the amount of spores per bee are 12.2% and 7.5%, 31.1%, and respectively 35.4% higher than those in the Shemonaikhinsky district are.

Notwithstanding that, it was impossible to determine statistically significant differences between dependence of prevalence of nosemosis and the apiary location, it is crucial to pay attention to the fact that in reality this value is higher in the mountainous areas than in the steppe. This is confirmed by other authors as well (*Dar & Sheikh Bilal, 2013*).

In addition, our research studies reliably prove, that the amount of spores per bee depends on the duration of the vegetation season: the shorter the vegetation season is, the more spores per bee are, and, consequently, the higher is the level of infestation of bees by nosemosis.

As per the authors of this research article, the orographic factors, geographical zone and duration of the vegetation season, defining the climate and flora of the area, predetermine the quantity and quality of feed, temperature, humidity and other conditions related to existence of the bees. Consequently, independently or together with with other factors, they impact infestation of bees with microsporidia, and are important markers to diagnose and forecast the distribution of nosemosis in any particular district. Nonetheless, in order to support these findings, it is required to have additional data collected under various climatic conditions and implementing different methods of beekeeping.

## CONCLUSIONS

The study results provided in this research article prove the distribution of the causative agent of nosemosis in honey bee in the East Kazakhstan Region. Predominantly, there is an average and high level of infestation recorded in bee families.

The total of apiaries affected by nosemosis in the mountainous areas is higher than in the lowlands. Still, there are no statistically significant differences with regard to occurrence of nosemosis and the apiary location.

The proportion of bee families affected by nosemosis at the apiaries also does not significantly depend on the natural and climatic conditions, or the orographic effects. Nonetheless, a close inverse correlation between the amount of spores per bee and the level of infestation of bee families with the duration of the vegetation season at the apiary location was defined. This provides grounds to assert that the environmental factors have an impact on formation and development of nosemosis.

The results of the research presented in the article indicate the need for further research aimed at increasing the number of studied apiaries, and above all the use of molecular biology methods to distinguish the species that cause nasal infection (PCR).

## ACKNOWLEDGEMENTS

We express our sincere gratitude to the beekeepers of Urzhar, Katon-Karagay, Shemonaikhinsky and Borodulikhinsky Districts of East Kazakhstan Region, who provided us with access to their bee families, as well as granted the bee samples.

### Funding

The authors received no funding for this work.

### Competing Interests

The authors declare there are no competing interests.

### Author Contributions

- Abdrakhman Baigazanov conceived and designed the experiments, performed the experiments, analyzed the data, prepared figures and/or tables, authored or reviewed drafts of the article, and approved the final draft.

- Yelena Tikhomirova conceived and designed the experiments, performed the experiments, analyzed the data, prepared figures and/or tables, authored or reviewed drafts of the article, and approved the final draft.
- Natalya Valitova conceived and designed the experiments, performed the experiments, prepared figures and/or tables, authored or reviewed drafts of the article, and approved the final draft.
- Maral Nurkenova analyzed the data, prepared figures and/or tables, authored or reviewed drafts of the article, and approved the final draft.
- Ainur Koigeldinova analyzed the data, prepared figures and/or tables, authored or reviewed drafts of the article, and approved the final draft.
- Elmira Abdullina analyzed the data, prepared figures and/or tables, and approved the final draft.
- Olga Zaikovskaya conceived and designed the experiments, analyzed the data, authored or reviewed drafts of the article, and approved the final draft.
- Nurgul Ikimbayeva conceived and designed the experiments, analyzed the data, authored or reviewed drafts of the article, and approved the final draft.
- Dinara Zainettinova analyzed the data, prepared figures and/or tables, and approved the final draft.
- Lyailya Bauzhanova analyzed the data, prepared figures and/or tables, and approved the final draft.

## Data Availability

The raw data is available in the Supplementary Files.

## Supplemental Information

Supplemental information for this article can be found online at http://dx.doi.org/10.7717/peerj.14430#supplemental-information.

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
