# Peer review of "Occurrence of Nosemosis in honey bee, Apis mellifera L. at the apiaries of East Kazakhstan"

_PeerJ, doi:10.7717/peerj.14430_

## Round 0.1 · original submission · Major Revisions

As the manuscript has been precisely reviewed by the experts in the disciplines and after thoroughly going through their reports it is clear that there are some shortcomings that need to be addressed before publishing it so the decision is recorded as a Major Revision.

Enclosed please find the reviewers’ report on your paper. One reviewer has minor recommendations for revision, the other has fairly substantial recommendations. Although their reports are very positive about your paper. All reviewers agree that you have a very promising idea but that serious revision is necessary, they also include helpful suggestions for improving the paper, especially regarding the language and grammar that need the attention of the authors. Because of the reviewers' concerns, I cannot accept the paper in its present form.

So, according to the recommendations made by the valuable reviewers, the manuscript needs comprehensive major revisions.

Reviewer 1 ·

Basic reporting

no commnet

Experimental design

no comment

Validity of the findings

no comment

Additional comments

the authors should pay more attention to the writing in this MS:
1. The scientific name should be in italics, such as "Apis mellifera" in line 61
2. In the section of "M&M", the introduction should be briefly, and some can be placed in the discussion.
3.in the section of "research method", Line 228,apis mellifera should be "A. mellifera"
4. line 238 and 239 should be deleted in an english version.
5. the references should be updated, only 3 articles were published since 2020

Reviewer 2 ·

Basic reporting

no comment

Experimental design

no comment

Validity of the findings

no comment

Additional comments

Annotated review Report

Title
Line 1: change title as below
Occurrence of Nosemosis in honey bee, Apis mellifera L. at the apiaries of East Kazakhstan
Abstract
1. Line 61: Apis mellifera must be etalic
2. Line 66-70 need grammar correction
3. Line 74-75 Not the clear statement like” natural and climatic
conditions”. Make this statement clear
4. At the end of abstract please suggestion for further or gape in present research may be added for further research
Introduction
1. Line 84 please add order and family name of bee species
2. Line 87-93 may be shifted in the start then come to Apis mellifera. Better is to write about the Apis mellifera , its utilization and importance in pollination. A lot of work has been done on this specific species, do concentrate on its instead of being general oh honey bees.
3. Line 107 add more recent references (Zander, 1909) and Nosema ceranae (Fries et al., 1996) as work has been done in Kashmir and many other countries recently
4. Line 130-141 need improvement of grammar
Materials & Methods

1. Convert 145-149 as below

The research was carried out at Agrotechnopark Scientific Center, at Veterinary and Food Safety
Laboratory of Shakarim State University of Semey. The apiaries of East Kazakhstan
Region of the Republic of Kazakhstan were used for sampling.

2. Line 156-169 there are a lot of facts and figures of the region but are without any reference. Please add references.

3. lines 156-207 are about the climate of the study region. It’s too much illustrated. Kindly make it shorter as it is not your methodological procedure or not any parameter of your research directly.

4. Line 213 change “through the years of 2018 to 2021’ to during 2018-21.

5. Change sentences at line 222-225
“Prior to laboratory testing, the living bees were frozen for immobilization at -20°C for 15-20 minutes (Methodological guidelines to diagnose nosemosis in honey bee, 1985; Grobov et al., 1987; Topolska I Hartwig, 2005; Ilyasov et al., 2013; Pohorecka et al., 2018; Shumkova et al., 2018)”
To
Prior to laboratory testing, live bees were frozen for immobilization at -20°C for 15-20 minutes by following the protocol set for Nosemosis diagnose in various studies ( author name missing 1985; Grobov et al., 1987; Topolska I Hartwig, 2005; Ilyasov et al., 2013; Pohorecka et al., 2018; Shumkova et al., 2018).

6. Change line Line-228-231
The analysis of infestation of honey bee (Apis mellifera L.) by microsporidia was carried out
pursuant to the Methodological guidelines for laboratory studies for nosemosis in honey bee,
officially approved by the Principal directorate of veterinary medicine of the USSR under the
Ministry of Agriculture dd. April 25, 1985 No. 115-6A
To
The analysis of infestation of honey bee (Apis mellifera L.) by microsporidia was carried out
pursuant to the methodological guidelines (described for laboratory studies of Nosemosis in honey bee) officially approved by the Principal directorate of veterinary medicine of the USSR under the Ministry of Agriculture dd. April 25, 1985 No. 115-6A.
7. Line 242-253 need attention of authors for language and grammar corrections
8. Line 265 -266 change whole sentence into two sentences and reduce the use of and
9. Line 267 give reference
Results

1. Line 280-81 Change the sentence “The conducted microscopic examination proves that nosemosis is widespread at the apiaries ofEast Kazakhstan Region”
To
Microscopic analysis indicated that nosemosis was widespread at the apiaries of
East Kazakhstan Region.
2. Line 284-286 change sentence “
In accordance with the data given in Table 2, it makes it possible to conclude that in the
mountainous regions (Katon-Karagay and Urzhar) infestation of the apiaries by nosemosis is higher than in the steppe (Borodulikhinsky and Shemonaikhinsky)”
To
According to results (Table 2), it is concluded that nosemosis infestation was higher in apiaries of mountainous regions (Katon-Karagay and Urzhar) as compared to the regions with steppe slope (Borodulikhinsky and Shemonaikhinsky).
3. Line 293-301 need rewrite up keeping in mind the grammer
4. 303 -304 change the sentences at line 303-304
“In accordance with the data given in Table 3, in Katon-Karagay District, no low level of infestation was detected among bee families.’
To
Low level of infestation was not detected among bee families of district Katon-Karagay district according to the results given in Table (30).

Discussion
1. Line 329, 333 please see your discussion at both these lines. These are confusing.
2. More recent reference are also available to discuss your results please add more discussion

Conclusions
Line 392 sentence not clear high level of infestation recorded in bee families (2 and 3 points).
Line 393-94
The total of apiaries affected by nosemosis in the mountainous areas is higher than in the
lowlands
Then it is clear that infestation was higher of Nosema apis
Please if possible mention species of Nosema or discuss nosema in your results and discussion part.
Line 402- 405 need rewrite up





References
This part need attention of authors for typological mistakes and formatting according to the journal format.

Table 2 at page 31
Total examined,
sites
These seems total examined colonies not sites

General comments
Manuscript is a great contribution in the field of Apidology. Language and grammar need attention of the authors. Species of Nosema if possible may be included to strengthen the results and discussion part. Major revision is suggested.

Reviewer 3 ·

Basic reporting

no comment

Experimental design

no comment

Validity of the findings

no comment

Additional comments

See the attached PDF

Annotated reviews are not available for download in order to protect the identity of reviewers who chose to remain anonymous.

---

## Round 0.2 · Minor Revisions

The manuscript was re-reviewed by the valuable experts, but still there are some shortcomings that need to be corrected in the manuscript before final submission. So minor revisions are recommended.

Reviewer 1 ·

Basic reporting

pass

Experimental design

pass

Validity of the findings

pass

Additional comments

no

Reviewer 2 ·

Basic reporting

no comment

Experimental design

no comment

Validity of the findings

no comment

Additional comments

Abstract
1.Line 66 t0 68 please correct grammar
2. Grammer of abstract still needs correction. Extensive use of and along with , may be avoided.
Introduction
Introduction
1. Line 87-91 , 89-101 Grammer of abstract still needs correction. Extensive use of and along with , may be avoided.


Results
1.
Line 320- 321 please rewrite it

From my side
Technically now the manuscript is in a good format but language of the manuscript (all sections) is still too poor and needs miner revision please.

---

## Round 0.3 · accepted · Accept

I have reviewed the manuscript thoroughly, all the comments are incorporated in the manuscript as mentioned by our valuable experts. Now the manuscript is okay for publication, and the decision is Accepted.